# Multi-Criteria Decision Approach to Design a Vacuum Infusion Process Layout Providing the Polymeric Composite Part Quality

**DOI:** 10.3390/polym14020313

**Published:** 2022-01-13

**Authors:** Sergey Shevtsov, Igor Zhilyaev, Shun-Hsyung Chang, Jiing-Kae Wu, Natalia Snezhina

**Affiliations:** 1Laboratory of Composite Materials and Structures, Southern Center of Russian Academy of Science, 344006 Rostov on Don, Russia; 2Institute of Polymer Engineering, University of Applied Sciences Northwestern Switzerland FHNW, CH-5210 Windisch, Switzerland; igor.zhilyaev@fhnw.ch; 3Dept. of Microelectronics, National Kaohsiung University of Science and Technology, Kaohsiung City 82445, Taiwan; shchang@nkust.edu.tw; 4Chen-Wei International Co., Ltd., Kaohsiung City 80764, Taiwan; jiingkae.wu@gmail.com; 5Dept. of Aircraft Engineering, Don State Technical University, 344000 Rostov on Don, Russia; snezhina_nataly@mail.ru

**Keywords:** polymeric composites, composite technology, vacuum infusion, process modeling and optimization, multi-criteria decision, Pareto frontier

## Abstract

The increasingly widespread use of vacuum assisted technologies in the manufacture of polymer-composite structures does not always provide the required product quality and repeatability. Deterioration of quality most often appears itself in the form of incomplete filling of the preform with resin as a result of the inner and outer dry spot formation, as well as due to premature gelation of the resin and blockage of the vacuum port. As experience shows, these undesirable phenomena are significantly dependent on the location of the resin and vacuum ports. This article presents a method for making a decision on the rational design of a process layout. It is based on early forecasting of its objectives in terms of quality and reliability when simulating its finite element model, on the correlation analysis of the preliminary and final quality assessments, as well as on the study of the cross-correlation of a group of early calculated sub-criteria. The effectiveness of the proposed method is demonstrated by the example of vacuum infusion of a 3D thin-walled structure of complex geometry.

## 1. Introduction

The processes of vacuum infusion in the production of polymer composite structures have gained significant popularity, especially in the last decade, due to their relative ease of implementation and inexpensive equipment [1,2,3,4,5,6]. They are increasingly used in aircraft, aerospace, shipbuilding and automotive industries. The implementation of vacuum assisted resin infusion technologies, as a rule, includes the following sequence of actions: laying-up of preform—dry fabric or glass or carbon fiber reinforcement on the mold surface; then covering of the whole preform by the flexible vacuum bag and sealing. The vacuum line connects to the preform through a vent (outlet), while atmospheric pressure acts on the resin inside the vessel, drawing it into the preform through the resin gate (inlet) and onto the vacuum bag, compacting the porous preform. A pressure gradient, which arises in the porous preform, causes the liquid resin to spread and fill the volume of the dry fabric. This process is accompanied by a group of related phenomena that are inherent to the vacuum infusion process. As the preform fills with resin, the average pressure gradient causing it to flow decreases. In addition, gradual resin curing, resulting in an increase in resin viscosity, also slows down the filling rate of the preform, the porosity of which is reduced due to compressive atmospheric pressure [7,8,9]. When several simultaneously moving streams occur, which is typical for infused structures of even relatively small complexity or in the presence of accelerating highly permeable (HPM) tissues or tapes, situations are possible when resin streams block empty areas, forming the inner or outer dry spots. Such streams can also block air access to the vacuum vent, stopping the movement of the resin [10,11,12,13]. The listed phenomena are extremely undesirable; their consequences are heterogeneity, deterioration of the mechanical properties of the composite in the body of the molded structure, which often leads to the impossibility of its correction and complete unsuitability. In addition, the complexity and high interdependence of the phenomena occurring in the infused structure, their high sensitivity to the process conditions, lead to instability of the achieved quality and bad repeatability of the final results. All most important problems of the vacuum infusion process implementation are discussed in detail in [14,15,16,17].

The significant labor intensity and cost of expensive components required to improve the quality indicators of the process by trial and error, gave rise to a large number of works oriented to the development of the computer modeling of the vacuum assisted resin infusion technologies. Most of these works [18,19,20,21,22] use the so-called FE / CV (Finite Element/Control volume) approach, which made it possible to overcome the fundamental difficulty of modeling processes with moving boundaries on which conditions change at each time step. Subsequently, some authors have proposed other approaches. Among them, the approach [23], which is based on the use of a lumped model that considers the vacuum infusion process as a dynamic system and significantly reduces the computational complexity, as well as approaches aimed at increasing the accuracy of reconstruction of the moving resin front using the level set [24] and the phase field equations [25,26]. Due to the fact that the process of vacuum infusion includes several interacting phenomena of different nature, all the numerical modeling methods require many experimental data obtained by independent, rather sophisticated methods, and empirical dependences between the process parameters built on the basis of these data. These are the dependences of the compaction and porosity of the preform on external pressure, its permeability on porosity, the dependence of the degree of cure and resin viscosity on time and temperature, and the dependence of the thermal properties of the preform at various stages of its filling with resin. The experimental technique and the results described in [8,9,27,28,29,30,31,32,33,34,35,36] are used in the modified empirical models of this work, whose construction is described in detail in [26,37].

The goal of most of the developed models of the vacuum infusion process is to understand the evolving dynamics of the formation of a resin flow pattern in a porous preform. Only a small part of the developed models was used in algorithms for inverse problems of optimization of quality [38,39,40,41] and (or) process productivity [42,43], and also to accept the tradeoff between quality and cost [44,45]. Two classes of parameters are most often used as the design variables: parameters of the process layout (number and location of injection gates and vacuum vents, their throughput) and process modes (temperatures of injected resin and preform heating, pressure in the vacuum line and in the resin injection gate). It depends on the computational complexity of the forward problem simulating the filling the preform with resin and on the number of numerical experiments that need to be carried out to find the region of the global or local optimum, taking into account the existing constraints. Therefore, in the studies on optimization of vacuum infusion processes, very simple geometries of composite preforms are considered.

The aim of the presented work is to significantly reduce the computational costs for solving the inverse problem of optimizing vacuum infusion of a rather complex composite part when varying the parameters of the process layout to obtain their values that provide the best attainable criteria for the quality and reliability. As the quality criterion of infusion is taken the unfilled volume of the preform at the time of the cessation of the resin spreading. To determine the auxiliary criteria that make it possible to predict the results of the process at the early stages of its simulation, an analysis of their correlation with the final values of the quality and reliability criteria is carried out, as well as an analysis of cross-correlation of these sub-criteria. The used approach follows predictive modeling technique, which is defined as the process of applying a model or mining algorithm to data in order to predict new or future observations [46]. This definition includes temporal prediction, in which observations up to time t are used to predict future values at time t1 > t. A simple statistical analysis of numerous observations to establish a relationship between the current and future state of the system is most relevant when it is impossible to analytically express such a relationship [47,48].

The remainder of this article is organized as follows. The second section contains a brief description of the forward modeling problem formulation with the most important modifications of the empirical dependences of the system parameters. In the third section, an example of computer modeling of a transient process of vacuum infusion of a 3D composite structure is considered, whose geometry is imported from a revised CAD model. As a result of the analysis of the time history of the predictive sub-criteria and the analysis of their correlation with the main quality objective—the relative voids volume *V_v_* in the preform when the resin stops, the combined predictive criterion is determined. The fourth section is focused on an analysis of optimum regions in criteria and design spaces in respect that the studied optimization problem is multi-objective with constraints. The final section is devoted to a discussion of the capabilities of the developed method and software tools, as well as the prospects for its perfectioning to improve the quality and productivity indicators achieved at the post-infusion stage due to controlled exposure to temperature and external pressure.

## 2. The Forward Modeling Problem Statement

The model under consideration describes the processes in a porous composite preform, which is a relatively thin-walled extended structure with a possible variation in the wall thickness, laid on an open mold. The example presented in the article does not take into account the layered structure, anisotropy of porosity and permeability of the preform. The developed software tool for modeling the forward problem is able to take into account the tensorial nature of the preform permeability, as well as its layered structure. However, the vacuum infusion process is almost never used when molding high loaded composite structures with orthotropic symmetry of the material. In such cases, transversal isotropy is ensured by the corresponding stacking sequence of the unidirectional or fabric layers. As shown in experimental studies [9,11], with a small and almost unchanged wall thickness of the molded composite structure, in-plane permeability plays a decisive role in the infusion process. These considerations justify the assumptions made in the presented work.

An arbitrary number of resin gates and vacuum vents can be attached to any location on the infused preform. The spreading of a liquid resin under the action of the gradient of internal propulsive pressure occurs during its continuous curing and changes in viscosity, while the pressure distribution in the preform depends on its local filling with resin. A local compressive strain of the preform under the action of the difference between external and internal pressure and, as a consequence, its porosity and permeability depend on its state (dry or wet). The evolution of the moving resin state is described by the equation of convection/thermal kinetics/diffusion of the degree of cure. All free surfaces of vacuum bag, which covers the preform, and open mold are exposed to convective action of ambient air at a controlled temperature. The system of the governing equations describes the entire complex of the listed phenomena in the overall system “filling a porous preform-mold” before the start of resin gelation, which prevents its spread. All the assumptions made are described below in the text of the section.

The system of the governing equations of the forward modeling problem is presented in Equations (1a)–(1d).
$$\left\{ \begin{array}{ll} \partial \varphi /\partial t+u\cdot \nabla \varphi =\nabla \cdot \gamma \nabla G; & (1a)\\ \left(1-V_{f}\right)\cdot \frac{2}{\pi }\frac{\zeta }{1+{\left(\zeta p_{m}\right)}^{2}}\frac{\partial p_{m}}{\partial t}-\nabla \left(\frac{\left[K\right]}{\mu }\nabla p_{m}\right)=0; & (1b)\\ \partial \alpha /\partial t-\left(\left[K\right]/\mu \right)\cdot \nabla p\cdot \nabla \alpha -\nabla \cdot \left(c_{\alpha }\nabla \alpha \right)=F(\alpha ,t,T); & (1c)\\ \rho_{pr}C_{pr}\partial T/\partial T+\nabla \cdot \left(-k_{pr}\nabla T\right)=Q_{exo}; & (1d) \end{array}\right.$$

The dependent variable φ∈[−1;1] in the Cahn-Hilliard phase field Equation (1a) [49], which describes the motion of the boundary between void and filled preform areas defines a local resin filling *V_r_* in accordance with $V_{r}=\left(\varphi +1\right)/2\in \left[0;1\right]$. In this equation *G* is a chemical potential, *γ* is the phase’s mobility, and **u** is the superficial resin velocity.

The second Equation (1b) of the coupled problem describing the dynamics of the viscous resin in unsaturated porous medium under pressure $p_{m}$ is the Richards equation modified by van Genuchten [50], where *V_f_* is a fiber volume fraction, [*K*] is a permeability tensor, *μ* is resin viscosity, *ζ* is the reciprocal of a certain reference pressure, taken equal to atmospheric one *ζ* = 1/*p_atm_*.

The convection/diffusion/kinetic Equation (1c) for the time-space evolution of degree of cure α combines three most important phenomena in the moving epoxy resin—diffusion of α, which depends on the kinetic and rheological state of the resin [51] and its displacement during resin spreading [11]. In this equation c_α_ is the diffusion coefficient, whereas source term *F(α*,*t*,*T)* describing the time dependence of the evolved α obeys the autocatalytic equation, which solution α(t) allows to determine an intensity of the exothermal heat source *Q_exo_*
(2)$$Q_{exo}=Q_{tot}\rho_{r}\left(1-V_{f}\right)\cdot V_{r}\cdot \partial \alpha /\partial t,$$
where *Q_tot_* is the total amount of heat released during the curing of the unit mass of the resin and *ρ_r_* is its mass density. The thermal properties of the preform: mass density $\rho_{pr}$, specific heat capacity $C_{pr}$ and thermal conductivity $k_{pr}$ are determined using mixing rule by using the same thermal properties of resin, dry preform and a local distribution of the resin filling *V_r_* [52]. All the initial and boundary conditions for Equations (1a)–(1d), as well as the dependencies of the fiber volume fraction *V_f_* on the compression pressure $p_{comp}(r,t)=p_{atm}-p_{m}(r,t)$, the permeability *K* (adopted as isotropic) of the preform on the porosity $V_{\varphi }$, thermal properties of resin and air on the temperature *T* and pressure *p_m_*, in detail are described in our work [26]. Note, however, that there have been significant changes to some of the system property dependencies.

For the dependence of the viscosity of the resin μ(T,α,t) on the degree of cure, instead of the Castro-Makosko model [16,29], the model
(3)$$\mu \left(T,\alpha ,t\right)=\mu_{0}(T^{in})\cdot \exp\left(\upsilon_{1}\cdot (T(t)-T^{in})+\upsilon_{2}\cdot \alpha (T,T^{in},t)\right)$$
was proposed. It is devoid of discontinuity at the resin gelation, it contains a directly measurable parameter—the viscosity $\mu_{0}(T^{in})$ at the initial temperature and only two coefficients $\upsilon_{1},\upsilon_{2}$ that are simply determined in the experiment. We use the refined semi-empirical dependence for the diffusion coefficient in Equation (1c), taking into account its decrease with increasing resin viscosity and dependence on the resin filling *V_r_*:(4)$$c_{\alpha }=c_{\alpha }^{0}\cdot \frac{\left(1+\tanh\left(\left(\alpha -1\right)/\sigma_{\alpha }\right)\right)}{\max\left(\log10\left(\mu \right),2\right)-1}V_{r},$$
where the values $c_{\alpha }^{0}=0.001$ and $\sigma_{\alpha }=0.15$ were determined on the basis of the compared results of experiments and numerical simulations carried out using a simple 1D system for molding a liquid epoxy resin. The global problem of prediction the dependence of the macroscopic properties of polymers on their properties at the molecular level (variations in inter- and intramolecular chemical reactivity, diffusivity, segmental compositions, etc.) was studied in [53] on the base of combined kinetic Monte Carlo and molecular dynamics simulations. This work presents a detailed analysis of the difficulties and restrictions encountered by even the sophisticated experimental methods, as well as theoretical methodologies, in particular, the limitations of their adequate description of the time evolution of the concentrations of reagents, intermediates and product concentrations before the onset of gelation. The approach implemented in the presented article is, by definition, phenomenological, i.e., excluding processes at the molecular level from consideration. Therefore, relation (4) should be considered as an empirical one, giving a correct qualitative description, substantiated experimentally, but the quantitative values of its parameters should be refined for each epoxy resin used.

The approach to solving the forward modeling problem formulated here is used in the next section in relation to the process of vacuum infusion of a 3D composite structure in order to identify the most important features of the process and find particular pre-calculated criteria that allow predicting its final indicators. All numerical values of the modeled system parameters are used from the manufacturer, from the reliable results of the referenced studies and own experiments, whose data are published in [25,26,37].

## 3. Modeled Vacuum Infusion System, Some Simulation Results and Predictive Criteria

The numerical experiments described below were carried out with a model of a composite preform laid on an open mold made of polymerized CFRP (Carbon Fibers Reinforced Plastic) with a thickness of about 5 mm. After eliminating the joints between the patches of the CAD (Computer Aided Design) model surface and attaching an open form to it (see Figure 1), the assembly geometry was imported into the CAE (Computer Aided Engineering) system Comsol Multiphysics 5.5 (Comsol LLC, Burlington, MA, USA) and subjected to finite element (FE) meshing.

A high-permeability (HPM) tape is laid along the perimeter of the preform, which has a gap at the location of the vacuum vent (see Figure 2). Its shift relative to a certain axis of the part and the gap between the edges of the HPM tape can vary.

The location of the resin injection gates did not change in the present study. It was accepted on the basis of the results of [26] as the most efficient for the spreading the two resin streams outgoing from them. Toray ER450 resin is introduced into the preform at an initial temperature of 70 °C, then the system is gradually heated for 15 min by convective heat flows and then maintained at an isothermal temperature. Numerical experiments were carried out at holding temperatures $T^{hold}$ of 80, 82 and 84 °C. Numerous simulations were performed at the varying temperatures, vacuum vent offset and vent-HPM gap. The simulated process took 4 h and was always longer than the time until the resin stopped completely. The aim of the study was to understand the patterns of the process, to determine its parameters that can be used to predict at the early stages of modeling its final results—quality, reliability and productivity. Some examples of the evolutions of these process parameters at the fixed vacuum vent offset and gap for three temperatures $T^{hold}$ studied are shown in Figure 3. In these plots the <…> symbol indicates the averaging of a certain value over the preform volume filled with resin by more than 10%.

In addition to the acceleration of processes with an increase in the temperature of isothermal holding, the following important regularities should be noted, which are undesirable for achieving the quality of the process. The graphs in Figure 3c show a sharp increase in viscosity near the vacuum port long before the moment the viscosity is equalized in the entire volume of the preform, which can lead to a slowdown in the resin flow (see Figure 3d) and even to blockage of the outlet. The initial increase in fiber content, caused by the action of the compressive pressure, gradually stabilizes and then decreases below the value before the start of the infusion, which is caused by the increase in the internal pressure in the preform when it is filled with resin (see Figure 3d). This phenomenon is highly undesirable for the strength of the molded structure, which was thoroughly studied in [54], where an improvement of the process using additional external pressure is proposed. However, the modification of the presented modeling method in accordance with this improvement is a task for future research.

Solving the problem of reducing the computation time to the moment when the results of the process can be predicted reliably is very important for using the described process simulation method for this process optimization. This task includes two components: finding the particular criteria that strongly correlate with the final quality indicator and determining the earliest possible moment in time for such a prediction. Such time instants must satisfy the requirement of being able to be clearly identified in the simulation. As a result of the research, it was found that the requirements for reliability and identification accuracy are satisfied when the maximum relative void volume of the preform becomes less than 0.1, i.e., when the filling level of 0.9 leaves the preform through the vacuum port (see Figure 4). Everywhere below, this time instant is denoted as *t*09.

It was noted above that some works [42,43] solve the problem of increasing the productivity of the process, i.e., reduce its duration to full completion, which is achieved after the resin is solidified throughout the entire volume of the preform. In this regard, it is of interest to compare the duration of the process until the moment *t*09 with its duration until the moment of stopping of the resin propagation, denoted hereinafter as *tstop*. The corresponding dependences of these durations on the parameters of the process layout are shown in Figure 5 in the form of stems diagrams. These dependences show that the duration *tstop* is less susceptible to the parameters of the process layout than *t*09. This result confirms the conclusion of the authors of [19,32,55] that believe that the post-infusion stage, which corresponds to the time interval after *tstop* and up to complete consolidation of the preform and determines the total process duration, should be considered separately for optimization of the process performance.

This conclusion, taking into account the capabilities of the formulated forward modeling problem, forces us to restrict ourselves to optimizing the quality criterion *V_v_*(*tstop*) of the process. To identify particular criteria, the values of which at the moment *t*09 will make it possible to predict the final value of the main quality criterion with the required accuracy and reliability, a comparative analysis of the response functions of these particular sub-criteria and the main criterion was carried out, as well as a correlation analysis of their interdependencies. The values of the main quality criterion and sub-criteria for each investigated temperature were obtained by simulating the process when changing the vacuum port shift and HPM gap with a step of 5 mm. Some of the results of such an analysis for the holding temperature $T^{hold}$ of 84 °C are presented in Figure 6 and Figure 7.

It is important to note that for all three investigated temperatures $T^{hold}$ of isothermal holding, the dependences presented in these figures have the same character, slightly differing in the values of the correlation coefficients. Therefore, here the dependences are given only for a single temperature of 84 °C. The response surfaces in Figure 6 for the residual void volume of the preform have a very similar relief with a clearly manifested minimum, which confirms the necessary sensitivity of the process quality criteria *V_v_*(*tstop*) and *V_v_*(*t*09) to its layout. In addition, the graph in Figure 7a shows a very strong correlation between the *V_v_*(*tstop*) and *V_v_*(*t*09) values. However, Figure 7b–d show a weak correlation between *V_v_*(*tstop*) and such preliminary calculated parameters as <α(t09)>, <μ(t09)> and $<p_{m}\left(t09\right)>$. Meanwhile, the smaller values of each of these parameters facilitate a more intensive resin flow at the final phase of vacuum infusion, that is, increase the reliability of the process. This conclusion suggests the advisability of using one or more of the listed parameters together with the *V_v_*(*t*09) criterion for more reliable achievement of the better results of the process.

This joint use of the two sub-criteria is possible in two ways. The first one involves the formulation of a single objective functional, which includes all sub-criteria with some weights, and the subsequent determination of the global or local optimum of the resulting scalarized functional. Its visualization, especially in the case of two sub-criteria, is very convenient in the coordinates of the design variables. In cases where there is a single optimum of such a combined functional (which, unfortunately, is not often encountered in practice), it is taken as the optimal solution. This approach allows for the constraints on all sub-criteria and design variables to be taken into account. However, this approach is characterized by arbitrariness in the choice of weights for each sub-criterion, which can distort the real evolution of the optimized process.

The second approach considers two or more sub-criteria as independent, but each in its own way characterizes the efficiency and quality of the developing process. For a non-trivial optimization problem with multiple goals, there is no single solution that simultaneously optimizes each objective. In this situation, it is necessary to make optimal decisions when there are trade-offs between two or more conflicting objectives. Such a compromise decision can be made on the basis of a set of Pareto-optimal solutions, each of which is characterized by the fact that none of the objective functions can be improved in value without degrading some of the other objectives. In the case of bi-objective problems, their solution is visualized by the Pareto frontier, often named the tradeoff curve, which can be drawn at the objective plane. This approach to solving problems of multi-objective optimization, as well as fuzzy logic, simulated annealing and genetic algorithms, is extremely effective in solving problems of optimal synthesis of layouts and parameters of chemical reactions processes, when there are sets of objective criteria and controlled variables of relatively large dimension [56,57]. So, to ensure the minimum curing time and polymeric chain length dispersity in the reversible deactivation radical polymerization process, a variation of the parameters of the temperature cycle and molar amounts of reacting monomer is used [58].

In the presented work, both complementary approaches are used: optimization of the combined scalarized functional and reconstruction of the Pareto frontier.

In the absence of the required experimental information on the relative role of auxiliary sub-criteria in achieving the optimum of the main quality objective *V_v_*(*tstop*), the resin viscosity <μ(t09)>, determined by averaging over the preform volume at time t09, was chosen as the second candidate for its use in the combined quality-reliability criterion. This choice is due to the close cross-correlation of three predictive sub-criteria (see Figure 8), whose correlation dependences with the main quality criterion *V_v_*(*tstop*) are presented in Figure 7b–d. The combined quality-reliability criterion is proposed in the form of a product of the normalized partial predictive criteria $V_{v}\left(t09\right)$ and <μ(t09)>:(5)$$CObj\left(t09\right)=\frac{V_{v}\left(t09\right)}{mean\left(V_{v}\left(t09\right)\right)}\cdot \frac{<\mu \left(t09\right)>}{mean\left(<\mu \left(t09\right)>\right)},$$
where the mean(…) operation is a simple averaging over a set of performed test numerical experiments.

Based on the analysis, some of the results of which are presented above, the following conclusions can be drawn. The layout of the process (localization of the vacuum port) and the temperature of isothermal holding $T^{hold}$ significantly affect the filling, the preform with resin, changing the relief of the response function of the main quality criterion *V_v_*(*tstop*), the location and value of its optimum. The strong correlation of the criterion $V_{v}\left(t09\right)$ with the main criterion *V_v_*(*tstop*) allows it to be used together with the <μ(t09)> criterion for predictive process optimization. As follows from the analysis of the properties of the used resin, studied in detail in [26], its viscosity during isothermal heating begins to increase sharply when it reaches a value of 2 Pa*s, which may be the reason for stopping the advance of the resin along the preform. This fact can be used as a constraint when choosing options for rational process design. Multiple numerical experiments have shown that the use of predictive criteria of a process to obtain reliable estimates of its final results reduces the computation time from 25% to 45%, which makes the use of predictive criteria preferable in the process optimization system.

In the next section, a combined procedure for improving the vacuum infusion process is considered at the stage of modeling the spread of resin in a preform before its gelation. This procedure includes bi-objective optimization using a combined criterion (5) and making the decision on the rational design of the process layout based on the analysis of the Pareto set.

## 4. Finding a Quasi-Optimal Design for the Vacuum Infusion Process Layout

To present in the most demonstrative form the method and results of solving this multi-objective problem, the localization and throughputs of the resin injection gates are taken to be fixed, found in our previous work [26]. The variable design parameters will be the position of the vacuum port relative to the preform and the gap between it and the edges of the HPM tape. All illustrative materials are given for isothermal holding at a temperature of 82 °C. For each variant of the variable design parameters, upon completion of the forward problem solution, the values of the predictive criteria $V_{v}\left(t09\right)$, <μ(t09)>, CObj(t09) were generated, as well as the maximum resin viscosity in the vicinity of the vacuum vent $\max\left(\mu^{out}\left(t09\right)\right)$. To implement automatic simulation stop, a stop condition $\underset{\Omega }{\min\left(V_{r}\right)}<0.1$ is defined in the Component Couplings of the FE model, and this stop condition has been added to the Time Dependent Solver setting. The sequential loading of the specified parameter combinations included in the list of the model input parameters is carried out by the solver in the Parametric Sweep mode with the input data and results being saved in a text file for further processing. With such a multiple sequential call of the forward modeling problem, it is very important to achieve the experimental objectives in the simplest manner with the minimum number of measurements and the least expense. The chosen strategy consists in carrying out the first numerical experiments at the extreme values (maximum and minimum setting) of the range of controlled variables with subsequent narrowing and displacement of the center of the numerical experiment plan (see Figure 9). The values $mean\left(V_{v}\left(t09\right)\right)$ and mean(<μ(t09)>) obtained as a result of averaging the sub-criteria, calculated at 9 points of the first plan, are used in the future when calculating the combined criterion CObj(t09) for all the following results of numerical experiments.

To obtain the results of bi-objective optimization in the most understandable form and to make a decision on the choice of a process layout, the simulation results of a limited number of variants were smoothed by third-order splines and presented in the form of 3D response surfaces and level line maps (see Figure 10 and Figure 11). Dependences of predictive criteria $V_{v}\left(t09\right)$ and <μ(t09)>, shown in Figure 10 and Figure 11, after normalization were used to build the dependence of the combined quality-reliability criterion CObj(t09) (see Figure 12).

As noted above, when the local viscosity reaches 2 Pa*s, a sharp increase in viscosity begins in the adjacent area. Such a situation near the vacuum port is likely to lead to its blockage, which is unacceptable and must be prevented by introducing an appropriate constraint. For the calculated variants, the smoothed response function $\max\left(\mu^{out}\left(t09\right)\right)$ is shown in Figure 13.

Outwardly the response functions <μ(t09)> and $\max\left(\mu^{out}\left(t09\right)\right)$ shown in Figure 11 and Figure 13 are similar, but the ranges of their variation are not comparable. The relationship between these sub-criteria, plotted using the tabulated values of their smoothed response functions, is shown in Figure 14. It shows that critical vacuum port blocking situations can occur even with a relatively low average resin viscosity, which is denoted as $<\mu \left(t09\right){>}_{block}$ in the plot. The sharp influence of sub-criterion $\max\left(\mu^{out}\left(t09\right)\right)$ on the dynamics of the vacuum infusion process forces to use this parameter as a constraint.

The results of the problem solved are represented in the decision space and in the quality-reliability criteria space in Figure 15 and Figure 16, respectively.

## 5. Discussion

The results of solving the problem, presented in two forms (see Figure 15 and Figure 16), show that such a problem cannot be an ideal optimization problem with a single optimal solution that is the best for all criteria without exception. The designer always has the freedom to make decisions. At the same time, the nature of the considered restrictions can be very different, and their composition is quite wide. Even in a simple problem statement presented in this article, when the objective sets include only two parameters, and only two design variables are taken into account, solving the problem of making the optimal decision is rather laborious. To reduce the computational complexity, it required the use of predictive criteria, whose effectiveness must be proven in a series of preliminary numerical and real experiments. The scope of the presented model is limited to the stage of infusion of the liquid resin, before its gelation, solidification and the achievement of the final mechanical properties. However, this stage is very important, since it is during its course that such quality indicators as the absence of dry spots and the minimum void volume of the porous preform are achieved. A useful feature of the developed modeling/optimization technology is the ability to analyze and reasonably select one of several Pareto-optimal solutions, which is illustrated in Figure 17, where screenshots of the distributions of viscosity and fiber volume fraction at times *t*09 and *tstop* of the solution corresponding to vacuum vent offset = 15 mm, HPM gap = 75 mm and $T^{hold}$ = 82 °C are shown. At these time instants the relative void volumes in the preform are *V_v_*(*t*09) = 0.0107 and *V_v_*(*tstop*) = 0.0044, that is very satisfactory result. Reducing the volume of voids by more than 2 times became possible due to the low viscosity, which provided the necessary flowing of the resin. However, it can be seen that the resin front at the stopping of the flow has shifted relative to the vacuum port surrounded by the high viscosity resin, which prevents a removing of the remaining air from the preform. This was due to the earlier arrival of the resin flow moving from the right to the vacuum vent. The deficiency can be eliminated by shifting the vent a short distance to the left.

A significant decrease in the fiber volume fraction, which is shown in Figure 17b, can also be eliminated only at the stage preceding the second stage of resin cure. Apparently, the best solution to this problem is the external controlled pressure technology proposed in [54]. Another important problem is related to the provision of the required thermal schedule during the implementation of the process by the out-autoclave method, when infrared irradiation of the preform can be effectively used [59]. The method requires thorough development for use in the technology of manufacturing composite structures of complex shape.

The above considerations substantiate the place and role of the proposed methods and approaches in the development of promising areas of modeling and practical implementation of varieties of vacuum infusion processes in the production of responsible polymeric composite structures. However, effective application of this technology is impossible in the absence of a large number of reliable experimental data on the properties of the used reinforcements and resins.

## 6. Conclusions

The article presents the formulation, methodology and results of solving the problem of making a reasonable decision about the parameters of the layout of vacuum infusion of a composite structure of complex shape. The proposed method for solving the forward modeling problem is based on the use of coupled equations of the phase field, Richards, convection/diffusion of the degree of cure of a moving liquid resin in a porous preform, heat transfer, taking into account the refined relations for the evolution of the rheological state of the resin and all system’s thermophysical properties during entire stage of filling the preform. The inherent accuracy of the model description of the front of a moving and continuously curing resin in a preform with varying porosity and permeability provides an effective reconstruction of the process dynamics and identification of such defects as non-impregnated dry spots, possible blockage of the vacuum vent, which result in a violation of the quality of the produced composite parts. These abilities are illustrated in the article using a simplified example of a homogeneous preform with quasi-isotropic permeability, depending on its local porosity, compressibility and resin filling. In order to use the developed method for solving the forward problem in process optimization systems by reducing the computation time, a group of predictive sub-criteria is proposed that are assessed long before the resin stops moving and provide a reasonable prediction of the final quality (residual unfilled volume of the preform) and process reliability at the end of the infusion stage. The result of solving the inverse problem is presented in two forms: in the decision space, allowing identifying the area of rational change in the design parameters of the process layout, considering the constraints, and as the Pareto set in the objectives space, identifying the best achievable results for each of the objectives. The strategy used to search for the region of the best of acceptable constrained decisions, which involves carrying out the first numerical experiments at the extreme values (maximum and minimum) of the range of controlled design variables, followed by narrowing and shifting the center of the numerical experiments plan, made it possible to obtain effective results after about 30 calls of the forward problem.

## Figures and Tables

**Figure 1 polymers-14-00313-f001:**
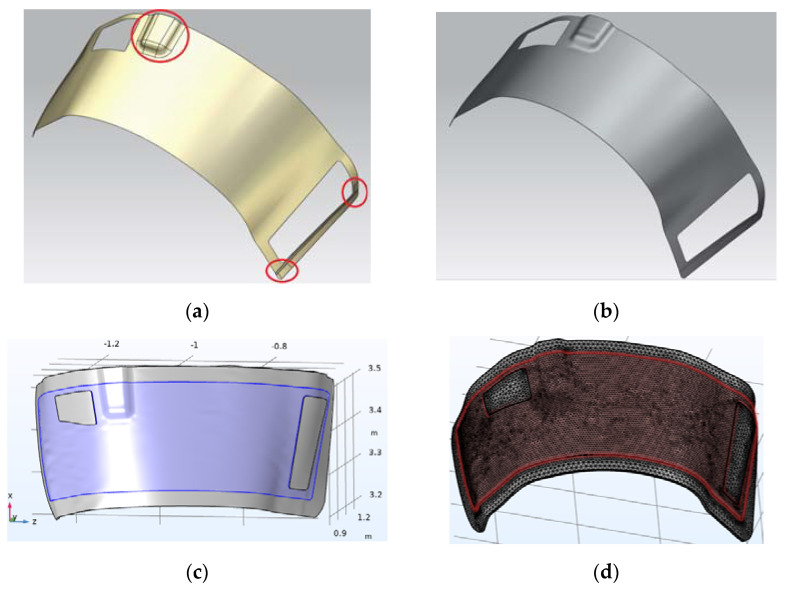
From the CAD model of the molded composite structure to the finite element (FE) model of the system of its vacuum infusion: (**a**) The initial CAD model; (**b**) CAD model of the structure after correction of its surface topology; (**c**) CAD model of assembled open mold with preform laid on it; (**d**) CAE model of the assembly after FE meshing.

**Figure 2 polymers-14-00313-f002:**
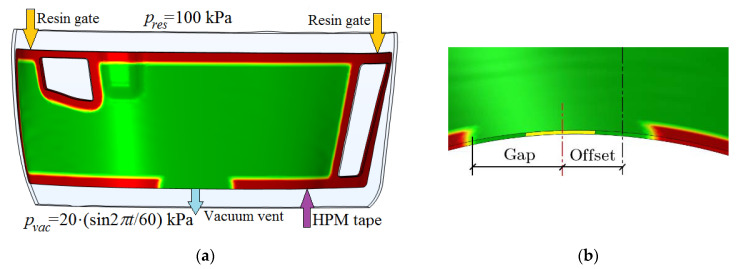
Layout of the vacuum infusion of the composite part: (**a**) HPM, resin injection gates and vacuum vent location on the preform; (**b**) The location of the vacuum vent relative to the axis divides the part into two approximately equal parts, and the distance from the vent to the edges of the HPM tape.

**Figure 3 polymers-14-00313-f003:**
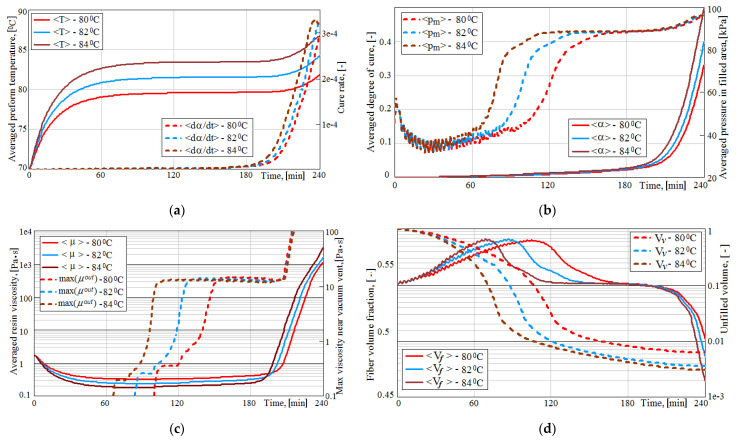
Superimposed time dependences of some process parameters: (**a**) Average temperatures <*T*> and resin cure rate <*dα/dt*>; (**b**) Average pressures <*p_m_*> and degree of cure <α>; (**c**) Average viscosity <*μ*> and maximum resin viscosity max(*μ*^out^) in the vicinity of the vacuum outlet; (**d**) Average fiber volume fraction <*V_f_*> and relative void volume *V_v_*.

**Figure 4 polymers-14-00313-f004:**
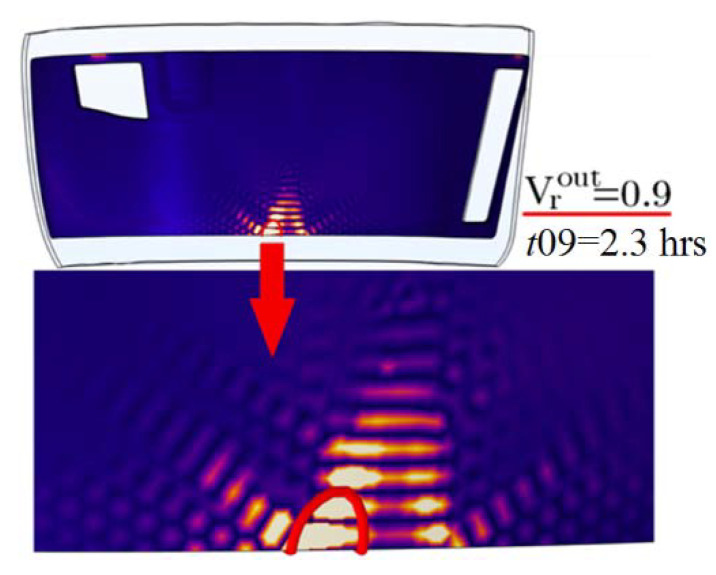
The position of the resin front with the 0.9 filling level just before leaving the outlet at the moment used for the most reliable prediction of the final results of the vacuum infusion process.

**Figure 5 polymers-14-00313-f005:**
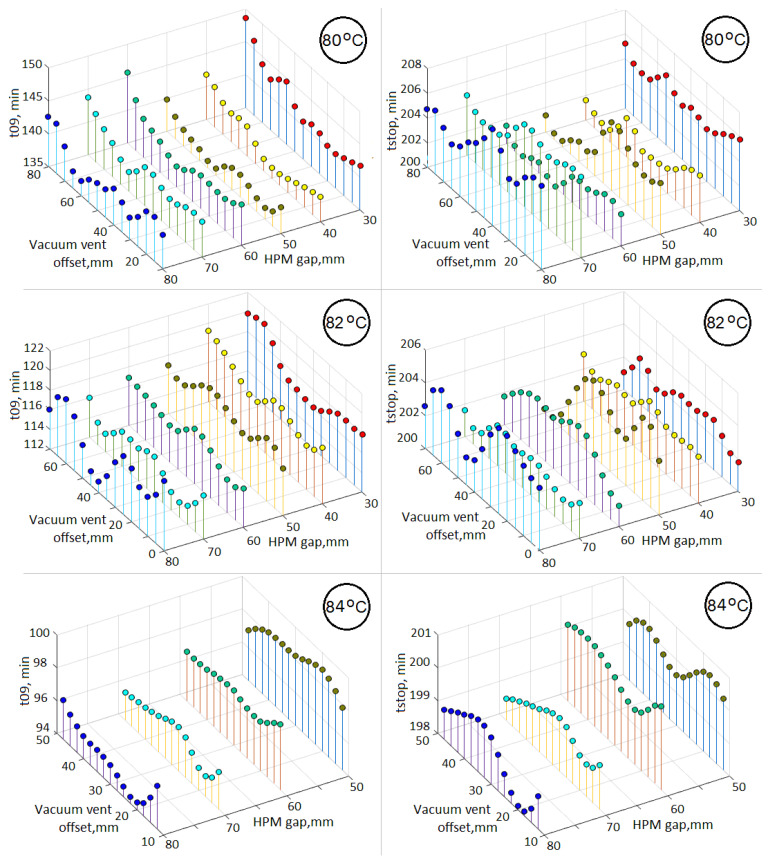
Dependencies of the process duration up to moments *t*09 (**left**) and *tstop* (**right**) at isothermal holding temperatures *T^hold^* of 80, 82, and 84 °C on the location of the vacuum vent.

**Figure 6 polymers-14-00313-f006:**
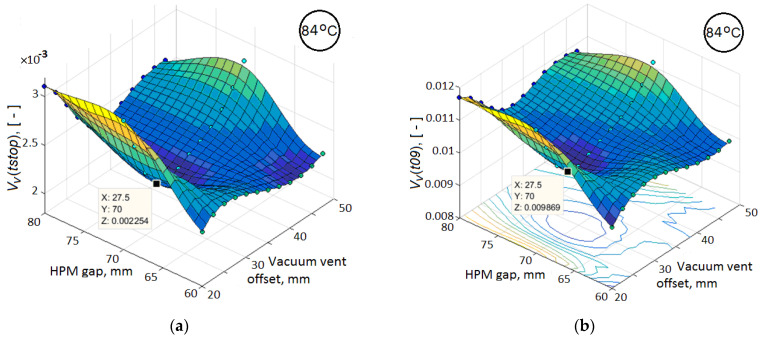
The response functions of the residual void volume *V_v_* in the preform on the parameters of the vacuum vent location: (**a**) at the time instant when motion of the resin stopped; (**b**) at the moment *t*09.

**Figure 7 polymers-14-00313-f007:**
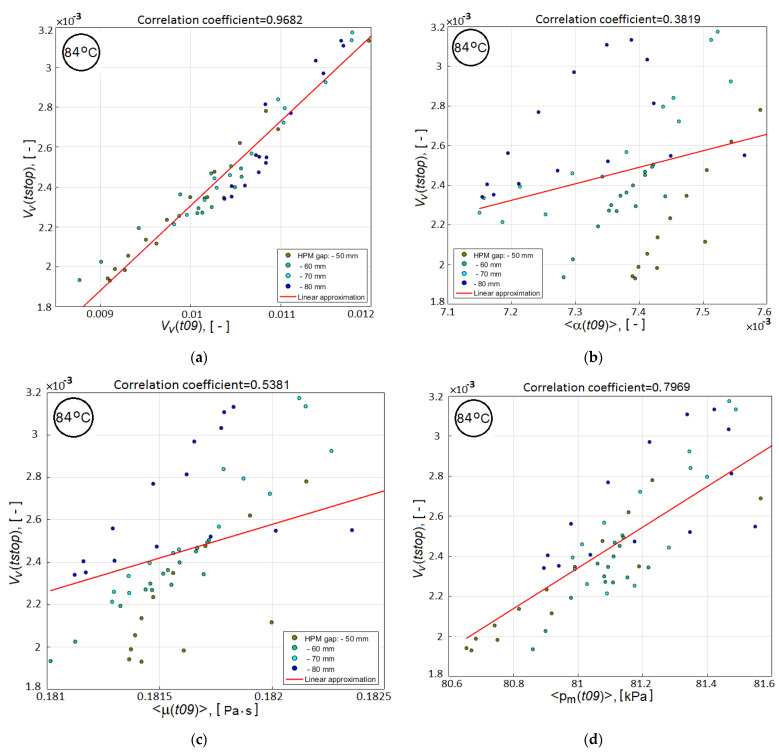
2D scatter plots showing the correlation between the main quality criterion at the end of the resin infusion and the sub-criteria, which are calculated at time instant *t*09: (**a**) relative void volume *V_v_*(*t*09); (**b**) averaged degree of cure <*α*(*t*09)>; (**c**) averaged viscosity <*μ*(*t*09)> and (**d**) averaged pressure <*P_m_*(*t*09)> in partially filled domain.

**Figure 8 polymers-14-00313-f008:**
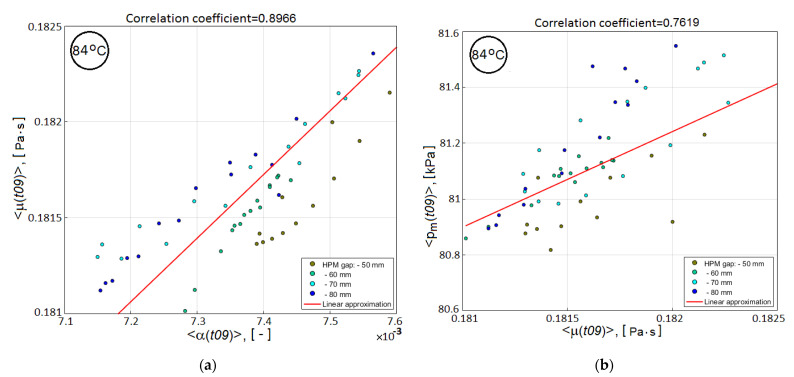
2D scatter plots showing the cross-correlation between the predictive sub-criteria: (**a**) <*μ*(*t*09)> and <*α*(*t*09)>; (**b**) <*P_m_*(*t*09)> and <*μ*(*t*09)>.

**Figure 9 polymers-14-00313-f009:**
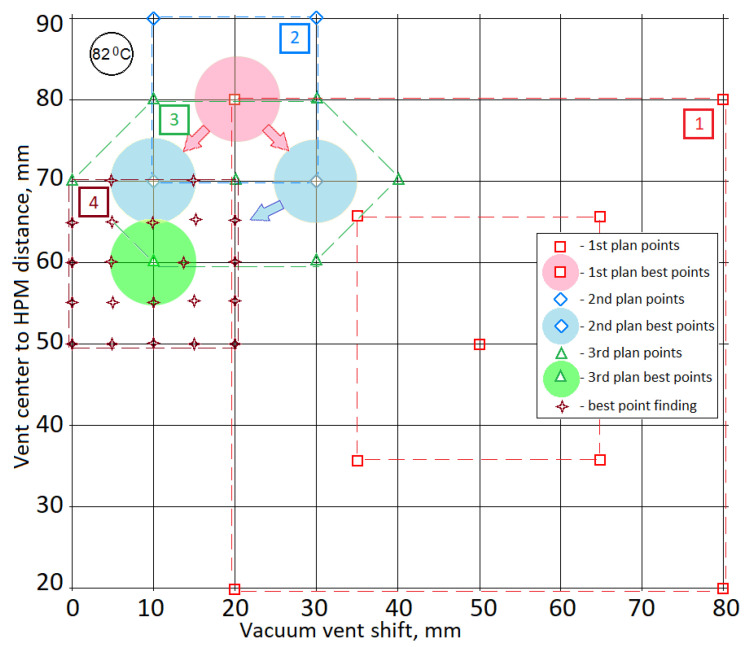
A sequence of plans for the performed numerical experiments to optimize the vacuum infusion process layout.

**Figure 10 polymers-14-00313-f010:**
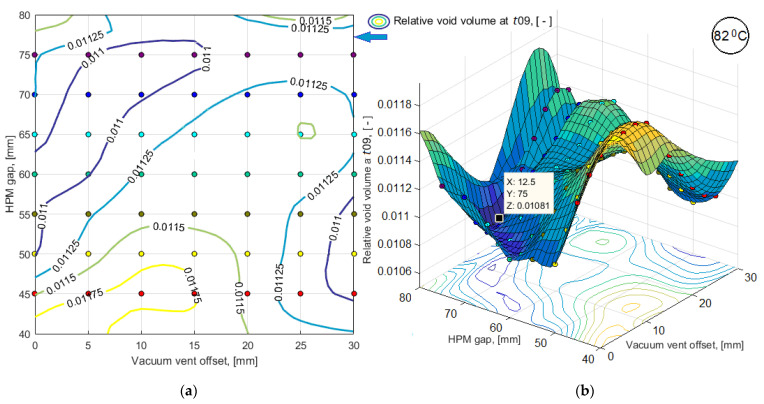
Results of simulation of the predictive sub-criterion *V_v_*(*t*09) response in the form of: (**a**) map of level lines; (**b**) smoothed 3D function.

**Figure 11 polymers-14-00313-f011:**
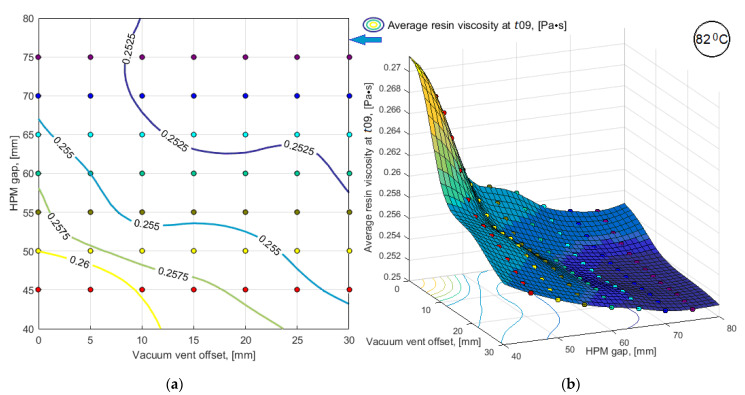
Results of simulation of the predictive sub-criterion <*μ*(*t*09)> response in the form of: (**a**) map of level lines; (**b**) smoothed 3D function.

**Figure 12 polymers-14-00313-f012:**
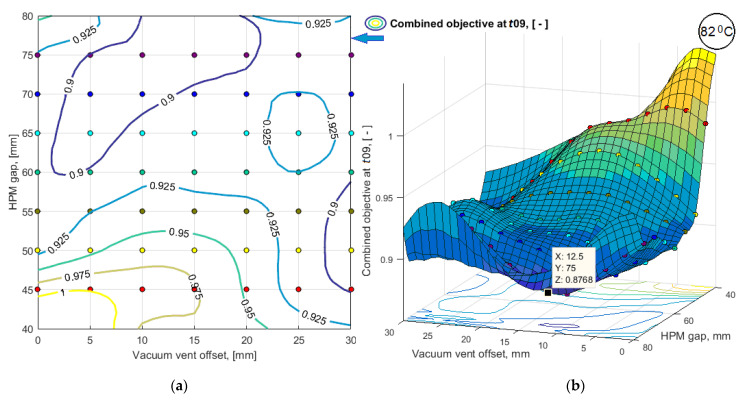
The response function of the predictive combined quality-reliability criterion *CObj*(*t*09) in the form of: (**a**) map of level lines; (**b**) smoothed 3D function.

**Figure 13 polymers-14-00313-f013:**
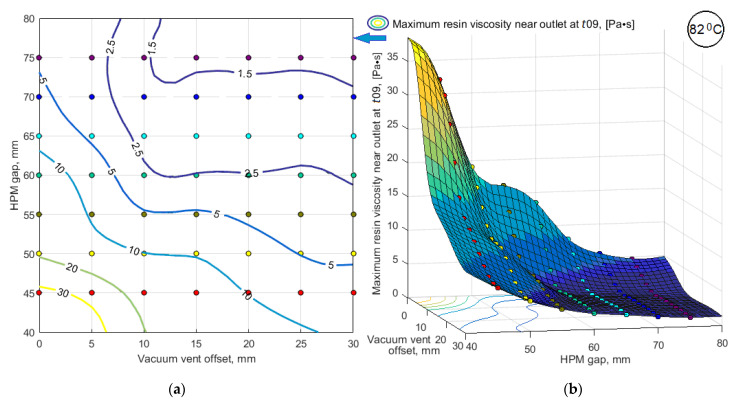
The response function of the predicted maximum resin viscosity in the vicinity of the vacuum vent max(*μ^out^*(*t*09)) in the form of: (**a**) map of level lines; (**b**) smoothed 3D function.

**Figure 14 polymers-14-00313-f014:**
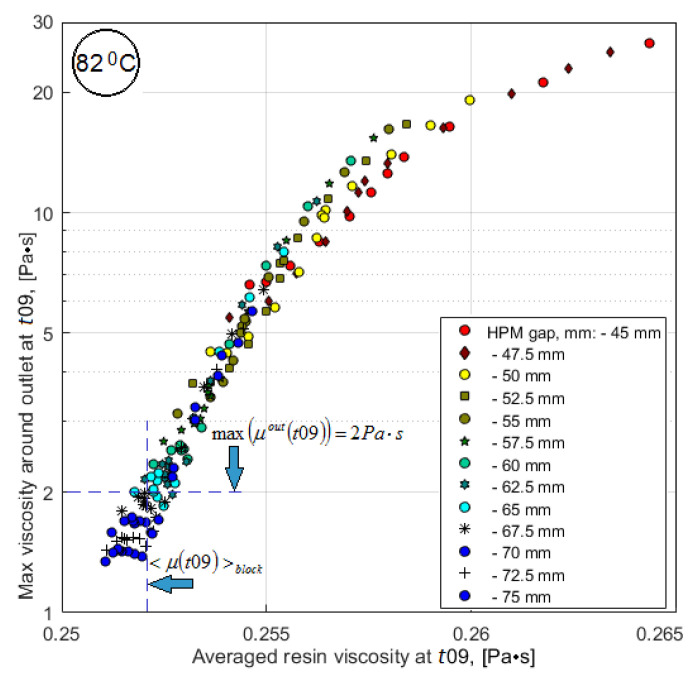
Interdependence between the averaged within the preform and maximum in the vicinity of the vacuum vent resin viscosities.

**Figure 15 polymers-14-00313-f015:**
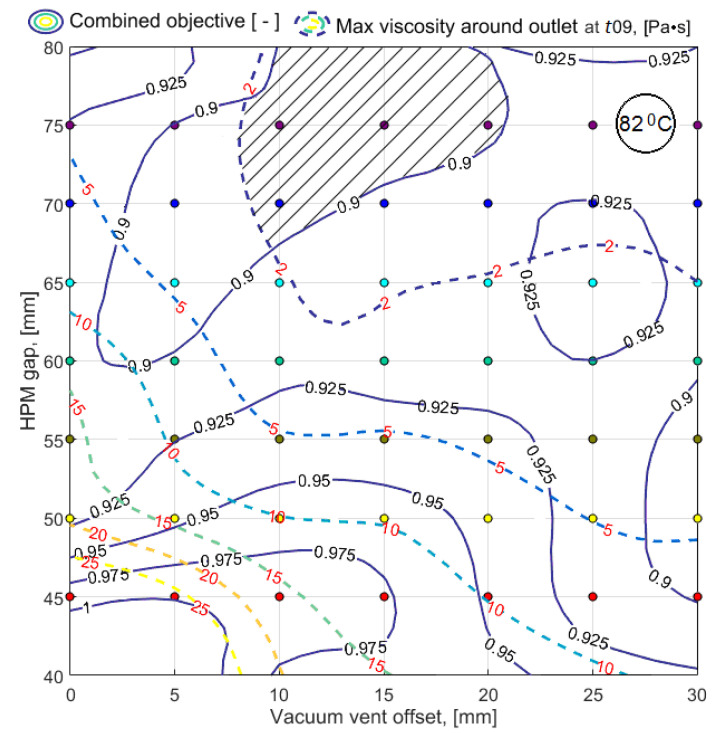
Map of the level lines of the combined quality-reliability criterion *CObj*(*t*09) (solid lines), superimposed with the lines of the constraint levels max(*μ^out^*(*t*09)) (dashed lines). The area of rational choice of the process layout parameters is highlighted by shading.

**Figure 16 polymers-14-00313-f016:**
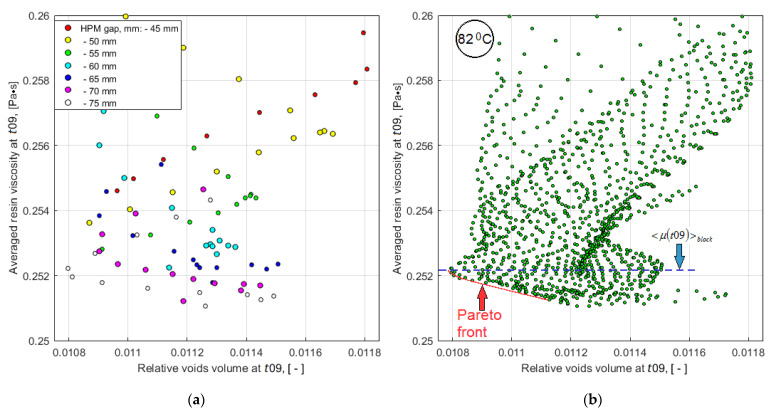
The scatter plots of the evaluating predictive criteria *V_v_*(*t*09) and <*μ*(*t*09)>: (**a**) The results of numerical experiments carried out according to the plans shown in Figure 9; (**b**) Results of tabulation of the smoothed response functions of each criterion with a step of 1 mm in both directions for the reconstruction of the Pareto front (solid red line).

**Figure 17 polymers-14-00313-f017:**
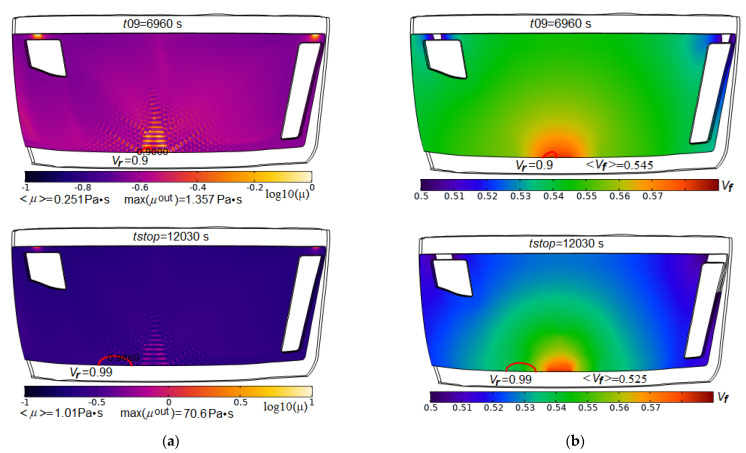
Screenshots of the spatial distributions for the logarithm of local resin viscosity (**a**) and fiber volume fraction (**b**) in partially filled domain at the time instants *t*09 and *tstop*. Red solid lines indicate the resin front positions with filling *V_r_* = 0.9 (upper pictures) and *V_r_* = 0.99 (lower pictures), respectively.

## Data Availability

Detailed information about the data confirming the results obtained in the course of the study can be requested from the authors of the article by e-mail.

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
