# Peer review of "Multi-Criteria Decision Approach to Design a Vacuum Infusion Process Layout Providing the Polymeric Composite Part Quality"

_polymers, 2022, doi:10.3390/polym14020313_

Round 1

Reviewer 1 Report

Line 107 to 144 need to separate from introduction. Maybe author can create new section on quality requirement ?. and also maybe can take out some paragraph in introduction for this new section.

Please check unit for temperature 'C. Some are missing.

Author also need to highlight the effect of viscosity of the resin towards the resin flowing. After all, this is very important point for the vacuum infusion successful.

Besides that, we we said about composite. We cannot run composite fabrication together with the reinforcement like glass fibre, natural fibre and other fillers. How about that modelling on the consequences of resin and weak points may occur ?

Author Response

Dear reviewer! All your comments and suggestions, which we find very useful and improve the content of our article, have been taken into account. Whenever possible, changes are displayed on a light green background.

  1. Line 107 to 144 need to separate from introduction. Maybe author can create new section on quality requirement ?. and also maybe can take out some paragraph in introduction for this new section.

Answer 1. Some of the text suggested by the reviewer has been removed. With this in mind, a small change was made to the description of the structure of the article (see lines 114-116). The necessary explanations contained in the removed part are given in section 3.

  1. Please check unit for temperature 'C. Some are missing.

Answer 2. All temperature units (degrees Celsius) have been corrected in the text and figures

  1. Author also need to highlight the effect of viscosity of the resin towards the resin flowing. After all, this is very important point for the vacuum infusion successful.

Answer 3. (lines 453-465, Fig.17) To demonstrate the effect of resin viscosity on flowability, we present two additional screenshots of the viscosity distribution in the preform at two times, t09 and tstop, when the resin is stopped. Their comparison shows that the maximum resin viscosity at t09 is no more than 1.4 Pa*s. This allows you to successfully complete the infusion process when the residual void volume reaches the minimum value 4.4e-3. These results are discussed and highlighted in the Discussion section.

  1. Besides that, we said about composite. We cannot run composite fabrication together with the reinforcement like glass fibre, natural fibre and other fillers. How about that modelling on the consequences of resin and weak points may occur ?

Answer 4. (lines 453-470, Fig.17) In the process of vacuum infusion of a porous preform, such disadvantages as depleted and re-enriched with binder areas, a reduced level of the fiber volume fraction can arise. They deteriorate the strength properties of the composite structure. We provide two additional screenshots showing the evolution of filling the porous structure with resin during the process. It's pretty uniform. However, a decrease in the fiber volume fraction, unfortunately, is characteristic of the infusion process. Elimination of this drawback is possible using the additional external pressure technology proposed in [54,55]. Its implementation is planned in our further studies. These results are discussed and highlighted in the Discussion section.

Reviewer 2 Report

The current work is very nice and I have only general comments on the framing of the work. The results are very convincing and I could not detect flaws there. I recommend publishing the work after following minor comments/suggestions:

L 112 Pareto. Please give some examples in the general polymer engineering field. Within Polymers one has Polymers 2015, 7, 655

“The example presented in the article does not take 148 into account the layered structure, anisotropy of porosity and permeability of the pre-149 form.” Please give the reader a feeling of the impact of these assumptions. Please also retake this in the conclusions.

“chapter” better “section” in this journal

Equation (4). Please mention that with more advanced models in the future the structure-property correlation (here viscosity vs curing blocks of the epoxy) can be improved. Pleas refer to Nat. Mater. 2021 20, 1422 in which such correlations are developed.

General comment: please add something about the reliability of the input parameters.

Author Response

Dear reviewer! All your comments are valid and the suggestions are very helpful. We used them to improve the readability and validity of the approaches and results presented in the article. Corresponding inserts are given in the text on a light blue background.

The current work is very nice and I have only general comments on the framing of the work. The results are very convincing and I could not detect flaws there. I recommend publishing the work after following minor comments/suggestions:

  1. L 112 Pareto. Please give some examples in the general polymer engineering field. Within Polymers one has Polymers 2015, 7, 655

Answer 1. (lines 329-338)

"... the Pareto frontier, often named the tradeoff curve, which can be drawn at the objective plane." This approach to solving problems of multiobjective optimization, as well as fuzzy logic, simulated annealing and genetic algorithms, is extremely effective in solving problems of optimal synthesis of layouts and parameters of chemical reactions processes, when there are sets of objective criteria and controlled variables of relatively large dimension [56,57]. So, to ensure the minimum curing time and polymeric chain length dispersity in the reversible deactivation radical polymerization process, a variation of the parameters of the temperature cycle and molar amounts of reacting monomer is used [58].

                In the presented work, both complementary approaches are used: optimization of the combined scalarized functional and reconstruction of the Pareto frontier.

[56] Massebeuf, S., Fonteix, C., Hoppe, S., Pla, F. Development of new concepts for the control of polymerization processes: Multiobjective optimization and decision engineering. I. Application to emulsion homopolymerization of styrene. J Appl Polym Sci 2003, 87, 2383–2396. DOI: 10.1002/app.12026.

[57] Mitra, K., Majumdar, S. Raha, S. Multiobjective dynamic optimization of a semi-batch epoxy polymerization process. Comput Chem Eng 2004, 28, 2583–2594. DOI: 10.1016/j.compchemeng.2004.07.003.

[58] Fierens, S.K., et al. Exploring the full potential of reversible deactivation radical polymerization using pareto-optimal fronts. Polymers 2015, 7(4), 655-679. DOI: 0.3390/polym7040655.

  1. The example presented in the article does not take 148 into account the layered structure, anisotropy of porosity and permeability of the pre-149 form.” Please give the reader a feeling of the impact of these assumptions. Please also retake this in the conclusions.

Answer 2.(lines 126-134) "The example presented in the article does not take into account the layered structure, anisotropy of porosity and permeability of the preform". The developed software tool for modeling the forward problem is able to take into account the tensorial nature of the preform permeability, as well as its layered structure. However, the vacuum infusion process is almost never used when molding high loaded composite structures with orthotropic symmetry of the material. In such cases, transversal isotropy is ensured by the corresponding stacking sequence of the unidirectional or fabric layers. As shown in experimental studies [9, 11], with a small and almost unchanged wall thickness of the molded composite structure, in-plane permeability plays a decisive role in the infusion process. These considerations justify the assumptions made in the presented work.

Insertion into Conclusions: (lines 496-498)

These abilities are illustrated in the article using a simplified example of a homogeneous preform with quasi-isotropic permeability, depending on its local porosity, compressibility and resin filling.

  1. “chapter” better “section” in this journal.

Answer 3. This change is made throughout the text.

  1. Equation (4). Please mention that with more advanced models in the future the structure-property correlation (here viscosity vs curing blocks of the epoxy) can be improved. Pleas refer to Nat. Mater. 2021 20, 1422 in which such correlations are developed.

Answer 4.(lines 188-200)

"...a simple 1D system for molding a liquid epoxy resin." The global problem of prediction the dependence of the macroscopic properties of polymers on their properties at the molecular level (variations in inter- and intramolecular chemical reactivity, diffusivity, segmental compositions, etc.) was studied in [53] on the base of combined kinetic Monte Carlo and molecular dynamics simulations. This work presents a detailed analysis of the difficulties and restrictions encountered by even the sophisticated experimental methods, as well as theoretical methodologies, in particular, the limitations of their adequate description of the time evolution of the concentrations of reagents, intermediates and product concentrations before the onset of gelation. The approach implemented in the presented article is, by definition, phenomenological, i.e. excluding processes at the molecular level from consideration. Therefore, relation (4) should be considered as an empirical one, giving a correct qualitative description, substantiated experimentally, but the quantitative values of its parameters should be refined for each epoxy resin used.

[53] De Keer, Lies, et al. Computational prediction of the molecular configuration of three-dimensional network polymers. Nat Mater 2021, 20(10), 1422-1430. DOI: 10.1038/s41563-021-01040-0.

  1. General comment: please add something about the reliability of the input parameters.

Answer 5. (lines 204-206)

All numerical values of the modeled system parameters are used from the manufacturer, from the reliable results of the referenced studies and own experiments, whose data are published in [25,26,37].

(lines 480-482)

However, effective application of this technology is impossible in the absence of a large number of reliable experimental data on the properties of the used reinforcements and resins.

Thank you very much for your suggestions.
